# Holographic Lens Resolution Using the Convolution Theorem

**DOI:** 10.3390/polym14245426

**Published:** 2022-12-11

**Authors:** Tomás Lloret, Marta Morales-Vidal, Víctor Navarro-Fuster, Manuel G. Ramírez, Augusto Beléndez, Inmaculada Pascual

**Affiliations:** 1Departamento de Óptica, Farmacología y Anatomía, Universidad de Alicante, Carretera San Vicente del Raspeig s/n, 03690 San Vicente del Raspeig, Spain; 2Instituto Universitario de Física Aplicada a las Ciencias y las Tecnologías, Universidad de Alicante, Carretera San Vicente del Raspeig s/n, 03690 San Vicente del Raspeig, Spain; 3Departamento de Física, Ingeniería de Sistemas y Teoría de la Señal, Universidad de Alicante, Carretera San Vicente del Raspeig s/n, 03690 San Vicente del Raspeig, Spain

**Keywords:** holographic lenses, resolution, convolution theorem, volume holography

## Abstract

The similarity between object and image of negative asymmetrical holographic lenses (HLs) stored in a low-toxicity photopolymer has been evaluated theoretically and experimentally. Asymmetrical experimental setups with negative focal lengths have been used to obtain HLs. For this purpose, the resolution of the HLs was calculated using the convolution theorem. A USAF 1951 test was used as an object and the impulse responses of the HLs, which in this case was the amplitude spread function (ASF), were obtained with two different methods: using a CCD sensor and a Hartmann Shack (HS) wavefront sensor. For a negative asymmetrically recorded HL a maximum resolution of 11.31 lp/mm was obtained. It was evaluated at 473 nm wavelength. A theoretical study of object-image similarity had carried out using the MSE (mean squared error) metric to evaluate the experimental results obtained quantitatively.

## 1. Introduction

Holography is a three-dimensional image reconstruction technique based on Gabor’s principle named *Wavefront reconstruction* [1]. It is an extremely intriguing approach that enables the information to be recorded in a recording medium and has advanced significantly as a result of enhancements made to holographic recording mediums as photopolymers [2].

Holographic optical elements (HOEs) are one of the most significant holographic applications [3,4]. HOEs are crucial because they can supersede complex, heavy, and curved refractive optical elements with a straightforward, lightweight, and flat element. Two coherent beams that are spatially overlapping each other form an interference pattern, which HOEs store. This pattern produces a photonic structure that can bend light in the desired direction.

Denisyuk first proposed the idea of a holographic mirror as a holographic HOE concept in 1962 [5]. With the advantages of achieving high optical power on a tiny substrate, a HOE may turn a hitting optical beam like a traditional lens [6]. Additionally, due to the element’s simplicity in coupling for any type of manipulation and the ability to multiplex two or more HOEs, a variety of functionalities can be gathered on a single substrate in accordance with its properties of high diffraction efficiency and narrow-band frequency [7,8]. Filters, credit cards, displays, couplers, projection systems, and storage all utilise HOEs as basic optical components [9,10,11,12,13]. Holographic lenses (HLs) have changed as several industries, including communications, photonics, and information processing, have progressed. HLs are now a component of optical imaging systems that are mostly used in head-mounted displays for virtual and augmented reality [14,15,16,17,18] or as non-image systems in light deflectors and concentrators [19,20,21,22,23,24].

In these applications, the optical and image quality of HLs is very important. For this purpose, some authors have studied the resolution of HLs using the modulation transfer function (MTF) [25,26], the Fourier transform [27,28] or the study of some quality metrics [29,30]. The MTF does not provide complete information on the HLs resolution, as it only provides information on the cut-off frequency in a region of the image. For that reason, studying the convolution of an object resolution test with the impulse response of the HLs is a good option to not lose part of the information.

The main points of this applicability are the new recording materials, which can perform in the most specific situations and applications. The manufacturing of HOEs typically involves the use of a variety of materials, including silver halide emulsion [31,32], dichromate gelatine [33,34], photoresist [35,36], photorefractive [12,37], or photopolymer [38]. In 1969, Close et al., employed photopolymers for the first time as holographic optical components [39]. Since then, a wide variety of photopolymer compounds has been developed for optical applications [40]. This is primarily because of their adaptability in terms of composition and design, in addition to other intriguing qualities like self-processing ability [41], affordability, variable thickness, good dimensional stability, high energy sensitivity, sharp angular selectivity, wide dynamic range, and flexibility. The significance of photopolymers in this context is expanding quickly [42].

However, routinely used hydrophilic photopolymers also include gelatin binders, poly(vinyl alcohol), and similar monomers [43,44,45,46]. Certain of these photopolymers’ unfavourable traits include the toxicity of some of their constituent parts; for instance, acrylamide has a high propensity to cause cancer. Recent developments in photopolymers reduce this issue by using low-toxicity materials [47,48,49] instead of conventional solvents to enhance environmental compatibility [50,51]. Such materials also offer good recycling qualities. In order to serve as a holographic recording material in optical applications, we developed *Biophotopol*, a low-toxicity photopolymer [52,53,54].

In this work, we have studied the resolution of HLs recorded on a low-toxicity recording photopolymer in asymmetrical recordings and with negative focal length. In previous papers we obtained the best quality results for negative asymmetric HLs [19,26,30]. We have obtained the convolution between the impulse response, obtained from a CCD sensor and an HS wavefront sensor, and a USAF object test. The resolution obtained directly experimentally with the CCD sensor has also been calculated.

## 2. Materials and Methods

### 2.1. Recording Material Composition

Volume phase transmission HLs were recorded in a Biophotopol photopolymer, a low-toxicity hydrophilic material made-up of one or more monomers in a binder, an electron donor, and a dye sensitizer. In earlier works, the component concentrations of Biophotopol have been thoroughly investigated [52,53,54] to make up optimal photopolymer layers with good optical responses for storing holographic lenses. Biophotopol composition was developed using sodium acrylate (NaAO) as the monomer (NaAO was generated in situ through a reaction of acrylic acid (HAO) with sodium hydroxide (NaOH) in a 1:1 proportion), water as a unique solvent, triethanolamine (TEA) serving as the initiator and plasticizer, riboflavin 5’-monophosphate sodium salt (RF) serves as a dye, and polyvinyl alcohol (PVA) as a binder (Mw = 130,000 g/mol, hydrolysis grade = 87.7%). All compounds were purchased from Sigma-Aldrich Quimica SL (Madrid, Spain). Figure 1 shows the Biophotopol chemical structures of the prepolymer components.

The optimized Biophotopol photopolymer’s component concentrations adequate to produce layers with the highest possible diffraction efficiency are shown in Table 1. The prepolymer solution was deposited in square glass moulds 6.5 × 6.5 cm${}^{2}$ (which have been previously washed and dried) by the force of gravity and left in dark inside an incubator (Climacell 111, Labexchange, Burladingen, Germany) for roughly 24 h at a controlled relative humidity, 60 ± 5% and a temperature = 20 ± 1 °C. The procedure was also carried out in a controlled light environment in which the photopolymer layer was not sensitive. During the drying period, some of the water evaporated until it reached equilibrium with the surrounding environment inside the incubator. An ultrasonic pulse-echo gauge (PosiTector 200, DeFelsko, Ogdensburg, NY, USA) was used to measure the physical thickness of the photopolymer dry layer obtaining a 200 μm thickness. The layers were now prepared for recording, which was started immediately.

We would like to point out that considering the capacity of TEA to form an H-bond with water, the low content of TEA made it more difficult for the NaAO to dissolve, stimulating crystallization during the drying process. The environmental conditions during the recording stage had to be controlled to avoid precipitation of NaAO on the surface of the photopolymer layer. Also, the permeability of PVA played an important role in the drying and recording stages. Therefore, the control of humidity, temperature, and TEA/water ratio in the dry layer with respect to the thickness and final composition of the photopolymer layer was very important to obtain high diffraction efficiency.

Not all of the dye in the photopolymer layer is used once the gratings are recorded, the dye remains in the unexposed zones and if the hologram is subjected to incoherent light, the reaction continues in these unexposed zones provoking the grating loss. Removing the extra dye after recording the grating is one approach to prevent this light response [55]. As a result, we exposed the register photopolymer layer to a 13.5 W (875 lumens at 6500 K, Lexman) LED bulb for 20 min. Due to the tiny amount of dye that is left in the exposed areas after curing, the gating is not removed. Over time the grating becomes more stable since this process only affects the leftover dye in the unexposed zones. The cured gratings were kept in natural lighting with seasonal variations in temperature and relative humidity without problems.

### 2.2. Experimental Setups

In Figure 2 it can be seen the experimental holographic setup used to calculate the HLs resolution in a direct experimental method. A 473 nm diode-pumped laser and a 633 nm He-Ne laser were used to illuminate the HLs. The beam, after being spatially filtered, was placed at the reconstruction angle $\theta_{C}$ that matched the recording angle [30]. The distance from the USAF test to the HL and the distance from the HL to the CCD sensor was 2$f_{HL}^{\prime }$ so that the magnification of the optical system M=1.

Table 2 shows the parameters of the HL reconstruction geometry. In addition, a typical experimental holographic setup for recording was used with the following parameters, recording wavelength 488 nm, object angle 0°, and reference angle 34.2° to obtain an HL focal length of 90 mm. More details such as impulse response using a CCD sensor and a Hartmann-Shack wavefront sensor can be found at [26,30].

### 2.3. Theoretical Background

#### 2.3.1. Convolution Theorem

Mathematically, a complex extended object can be represented as a weighted sum of impulse functions [56]. The impulse response of the HL studied is the ASF, which can be independent of the object plane position, in which case it is called invariant under translations. In addition, if there is no distortion in the system, the image plane coordinates are related linearly with the coordinates of the object plane through the lateral magnification *M*. Therefore, the image of an extended object can be calculated as an overlay of weighted ASF through the following direct operation.
(1)$$I\left(x,y\right)=\int \int O\left(u,v\right)\cdot ASF\left(u-\frac{x}{M},v-\frac{y}{m}\right)dudv$$
where O(u,v) and I(x,y) represent the object and the image, respectively. This integral is called *convolution*. Therefore, the image of a complex object can be seen as a convolution of that object and the impulsive response of the system.
(2)Image=Object⊛ASF

#### 2.3.2. Amplitude Spread Function (ASF)

Let us start by considering that in the exit pupil of an HLs there is a spherical wavefront. It can be described through the sagitta of this wavefront.

Considering the geometry of Figure 3 and using the Pythagorean theorem, it can be seen that(3)$${\left(R-z\right)}^{2}+h^{2}=R^{2}$$(4)$$-2zR+z^{2}+h^{2}=0$$

Considering small apertures $\left(Z<<\right)$, it can be considered $Z^{2}\approx 0$, so knowing that $h=\sqrt{x^{2}+y^{2}}$ it can be shown(5)$$z≃\frac{h^{2}}{2R}=\frac{\left(x_{p}^{2}+y_{p}^{2}\right)}{2R}$$

Working within the paraxial approximation, we can define the field in the HL exit pupil plane as a complex exponential function of the sagitta of the outgoing wavefront.(6)$$u\left(x_{p},y_{p}\right)=\exp\left[-ik\cdot Z\right]=\exp\left[-ik\cdot \frac{\left(x_{p}^{2}+y_{p}^{2}\right)}{2f^{\prime }}\right]$$

Generalising this to the case of an object point *O* and to any optical system, we have that the field is given as(7)$$u\left(x_{p},y_{p}\right)=\exp\left[-ik\cdot Z_{s^{\prime }}\right]=\exp\left[-ik\cdot \frac{\left(x_{p}^{2}+y_{p}^{2}\right)}{2s^{\prime }}\right]$$

Knowing that the size of the lenses is finite and the pupil is circular, we can write the field in the exit pupil plane as(8)$$u\left(x_{p},y_{p}\right)=p\left(x_{p},y_{p}\right)\cdot \exp\left[-ik\cdot \frac{\left(x_{p}^{2}+y_{p}^{2}\right)}{2s^{\prime }}\right]$$where$$p\left(x_{p},y_{p}\right)=p\left(r_{p}\right)=\left\{\begin{array}{ccc} 1 & \; & r_{p}\leq R_{exitpupil}\\ 0 & \; & r_{p}>R_{exitpupil} \end{array}\right.$$and $r_{p}$ is defined as $r_{p}=\sqrt{x_{p}^{2}+y_{p}^{2}}$.

However, the objective is to obtain the Intensity or Amplitude distribution in a given plane, usually the paraxial image plane of the exit pupil. The tool we have to propagate the field is the Fresnel integral. Therefore(9)$$u_{s^{\prime }}\left(x^{\prime },y^{\prime }\right)=A\cdot \int \int_{-\infty }^{\infty }u\left(x_{p},y_{p}\right)\cdot e^{i\pi \left(\frac{x_{p}^{2}+y_{p}^{2}}{\lambda s^{\prime }}\right)}\cdot e^{-i2\pi \left(\frac{x^{\prime }x_{p}^{2}+y^{\prime }y_{p}^{2}}{\lambda s^{\prime }}\right)}dx_{p}dy_{p}$$

Applying the definition of the Fourier transform we have that$$FT\left[h(x,y)\right]=H\left(u,v\right)=\int h\left(x,y\right)e^{i2\pi (ux+vy)}dxdy$$⇓(10)$$u_{s^{\prime }}\left(x^{\prime },y^{\prime }\right)=A\cdot FT{\left[u\left(x_{p},y_{p}\right)\cdot e^{i\pi \left(\frac{x_{p}^{2}+y_{p}^{2}}{\lambda s^{\prime }}\right)}\right]}_{u=\frac{x^{\prime }}{\lambda s^{\prime }},v=\frac{y^{\prime }}{\lambda s^{\prime }}}$$

Using the relationship obtained in Equation (Equation 8), and applying the definition of the wavenumber as $k=\frac{2\pi }{\lambda }$, it is obtained(11)$$u_{s^{\prime }}\left(x^{\prime },y^{\prime }\right)=A\cdot TF{\left[\left(p\left(x_{p},y_{p}\right)\cdot e^{-i\left(\frac{2\pi }{\lambda }\right)\left(\frac{x_{p}^{2}+y_{p}^{2}}{2s^{\prime }}\right)}\right)\cdot e^{i\pi \left(\frac{x_{p}^{2}+y_{p}^{2}}{\lambda s^{\prime }}\right)}\right]}_{u=\frac{x^{\prime }}{\lambda s^{\prime }},v=\frac{y^{\prime }}{\lambda s^{\prime }}}$$

Therefore, $u_{s^{\prime }}\left(x^{\prime },y^{\prime }\right)$ been simplified to(12)$$u_{s^{\prime }}\left(x^{\prime },y^{\prime }\right)=A\cdot FT{\left[p(x_{p},y_{p})\right]}_{u=\frac{x^{\prime }}{\lambda s^{\prime }},v=\frac{y^{\prime }}{\lambda s^{\prime }}}$$

Doing the change of variable $P\left(u,v\right)=FT\left[p(x_{p},y_{p})\right]$, it can be shown that P(u,v) can be described as(13)$$P\left(u,v\right)=2\frac{J_{1}\left(2\pi \sqrt{u^{2}+v^{2}}R_{ps}\right)}{2\pi \sqrt{u^{2}+v^{2}}R_{ps}}$$that represents the impulse response function of the system. The impulse response function describes the response of an imaging system to an object point and can be defined in amplitude or intensity. For an optical system working with coherent light, and whose pupil is circular, the impulse response is called the *Amplitude Spread Function* (ASF). On the other hand, when the light source is incoherent, this function is called the *Point Spread Function* (PSF), and represents the intensity distribution in the image plane. Moreover, for an aberration-free (diffraction-limited) system with a circular pupil, the PSF has an analytical solution and is the well-known *Airy’s Disc*. Therefore we conclude that, while for systems working with incoherent light the impulse response is defined in intensity, for systems working with coherent light it is defined in amplitude.**Amplitude**
**Intensity**
$P\left(u,v\right)=2\left(\frac{J_{1}\left(2\pi \sqrt{u^{2}+v^{2}}R_{ps}\right)}{2\pi \sqrt{u^{2}+v^{2}}R_{ps}}\right)$
$I\left(u,v\right)={\left|2\frac{J_{1}\left(2\pi \sqrt{u^{2}+v^{2}}R_{ps}\right)}{2\pi \sqrt{u^{2}+v^{2}}R_{ps}}\right|}^{2}$where $J_{1}$ represents a first order Bessel function.

To generalise this to the case where the emergent wavefront of the exit pupil is aberrated, we turn to the geometry of Figure 4. In the figure, it can be seen the relationship between the wave aberration (*W*) and the ray aberration (Δ(x,y)). In addition it can be seen that the ideal and real sagitta can be related as(14)$$Z_{real}=W+Z_{ideal}$$

Therefore, the real field $u_{real}\left(x_{p},y_{p}\right)$ is defined as(15)$$u_{real}\left(x_{p},y_{p}\right)=p\left(x_{p},y_{p}\right)\cdot e^{-ik\cdot Z_{real}}=p\left(x_{p},y_{p}\right)\cdot e^{-ik\cdot W}\cdot e^{-ik\cdot Z_{ideal}}$$which, as in the ideal case, we propagate by means of the Fresnel integral(16)$$u_{s^{\prime }}\left(x^{\prime },y^{\prime }\right)=A\cdot \int \int_{-\infty }^{\infty }u_{real}\left(x_{p},y_{p}\right)\cdot e^{i\pi \left(\frac{x_{p}^{2}+y_{p}^{2}}{\lambda s^{\prime }}\right)}\cdot e^{-i2\pi \left(\frac{x^{\prime }x_{p}^{2}+y^{\prime }y_{p}^{2}}{\lambda s^{\prime }}\right)}dx_{p}dy_{p}$$and if we now introduce $u_{real}$ obtained in (Equation 15), in the Equation (Equation 16) and considering (Equation 5) we obtain(17)$$u_{s^{\prime }}\left(x^{\prime },y^{\prime }\right)=A\cdot \int \int_{-\infty }^{\infty }p\left(x_{p},y_{p}\right)\cdot e^{-ik\cdot W}\cdot e^{-ik\cdot Z_{ideal}}e^{i\pi \left(\frac{x_{p}^{2}+y_{p}^{2}}{\lambda s^{\prime }}\right)}\cdot e^{-i2\pi \left(\frac{x^{\prime }x_{p}^{2}+y^{\prime }y_{p}^{2}}{\lambda s^{\prime }}\right)}$$

If, as before, we use the Fourier transform, we have that in the Gaussian image plane, the field is given as(18)$$u_{s^{\prime }}\left(x^{\prime },y^{\prime }\right)=A\cdot FT{\left[p\left(x_{p},y_{p}\right)\cdot e^{-ik\cdot W(x_{p},y_{p})}\right]}_{u=\frac{x^{\prime }}{\lambda s^{\prime }},v=\frac{y^{\prime }}{\lambda s^{\prime }}}$$and using the characteristic equation of the generalised pupil function(19)$$P\left(x_{p},y_{p}\right)=p\left(x_{p},y_{p}\right)\cdot e^{-ik\cdot W(x_{p},y_{p})}$$we have that the field in the image plane, with the presence of aberrations, is defined as(20)$$u_{s^{\prime }}\left(x^{\prime },y^{\prime }\right)=A\cdot FT{\left[P(x_{p},y_{p})\right]}_{u=\frac{x^{\prime }}{\lambda s^{\prime }},v=\frac{y^{\prime }}{\lambda s^{\prime }}}$$

In the case of an aberrated optical system, the impulse response can be defined as the Fourier transform of a complex function *P*, commonly referred to as the generalised pupil function. And we can further define the *Amplitude Transfer Function* (ATF) as the Fourier transform of the impulse response. This indicates that the ATF is proportional to the complex function *P*. It is evident that the bandwidth of the ATF is not affected by the presence of aberrations. It should be noted that the only effect of the aberrations is to introduce phase distortions into the bandpass.(21)$$ASF\left(x^{\prime },y^{\prime }\right)=A\cdot FT{\left[P(x_{p},y_{p})\right]}_{u=\frac{x^{\prime }}{\lambda s^{\prime }},v=\frac{y^{\prime }}{\lambda s^{\prime }}}$$(22)$$ATF\left(x^{\prime },y^{\prime }\right)=FT\left[ASF(x^{\prime },y^{\prime })\right]$$


#### 2.3.3. Case 1: ASF Obtained Using a CCD Sensor

In this case, to perform the convolution between the impulse response and an objective test, it is necessary to obtain the impulse response in complex amplitude, but this is not possible because when working with a CCD sensor only the impulse response in intensity can be obtained. A good option is to make an approximation and obtain the absolute value of the amplitude. According to [26], the intensity in the image plane can be calculated as:(23)$$I\left(x^{\prime },y^{\prime };z^{\prime }\right)=\frac{1}{B^{2}}{\left|\int \int_{S}A\left(x,y\right)exp\left[i\Delta (x,y;x^{\prime },y^{\prime };z^{\prime })\right]dxdy\right|}^{2}$$
and the amplitude approximation can be obtained as
(24)$$A\left(x^{\prime },y^{\prime };z^{\prime }\right)=\frac{1}{B}\left|\int \int_{S}A\left(x,y\right)exp\left[i\Delta (x,y;x^{\prime },y^{\prime };z^{\prime })\right]dxdy\right|$$
so ASF can be considered to be $ASF\left(x^{\prime },y^{\prime };z^{\prime }\right)\approx A\left(x^{\prime },y^{\prime };z^{\prime }\right)$.

#### 2.3.4. Case 2: ASF Obtained Using an HS Wavefront Sensor

On the other hand, the complex amplitude impulse response can be obtained using the Hartmann-Shack wavefront sensor. Aberrations can be defined in the exit pupil plane using the wavefront aberration function as
(25)$$W\left(\rho ,\theta \right)=\sum_{n=0,m=-n}^{k}\sum_{n-|m|=par}^{n}C_{n}^{m}\cdot Z_{n}^{m}\left(\rho ,\theta \right)$$
where $C_{n}^{m}$ are the Zernike coefficients which for negative asymmetric HL according to [26,30] is dominated by spherical aberration ($C_{0}^{4}$) and $Z_{n}^{m}\left(\rho ,\theta \right)$ is the general form given by
(26)$$Z_{n}^{m}\left(\rho ,\theta \right)=\left\{\begin{array}{ccc} N_{n}^{m}R_{n}^{\left|m\right|}\left(\rho \right)\cos\left(m\theta \right) & para & m\geq 0\\ -N_{n}^{m}R_{n}^{\left|m\right|}\left(\rho \right)\sin\left(m\theta \right) & para & m<0 \end{array}\right.$$
where the superscript *m* denotes the angular frequency, and the subscript *n* denotes the degree of the radial polynomial. In addition, the explicit form of the radial polynomial, $R_{n}^{\left|m\right|}$, is defined as
(27)$$R_{n}^{\left|m\right|}\left(\rho \right)=\sum_{s=0}^{(n-|m|)/2}\frac{{\left(-1\right)}^{s}\left(n-s\right)!}{s![0.5(n+|m|)-s]![0.5(n-|m|)-s]!}\rho^{n-2s}$$
which is a polynomial of degree *n* containing the terms $\rho_{n}$, $\rho_{n-2}$,⋯, $\rho_{m}$. It can also be deduced that the normalization factor, $N_{n}^{m}$ , is defined as
(28)$$N_{n}^{m}=\sqrt{\frac{2(n+1)}{1+\delta_{m0}}}$$
where $\delta_{m0}$ is the Kronecker delta.

Finally, the ASF is given as
(29)$$ASF\left(x^{\prime },y^{\prime };z^{\prime }\right)=A\cdot FT{\left[p\left(x^{\prime },y^{\prime }\right)\cdot e^{-ik\cdot W(x^{\prime },y^{\prime })}\right]}_{u=\frac{x^{\prime }}{\lambda f_{HL}^{\prime }},v=\frac{y^{\prime }}{\lambda f_{HL}^{\prime }}}$$
where $x^{\prime }=r\cos\theta$ and $y^{\prime }=r\sin\theta$.

### 2.4. Metric Based on the Similarity between Object and Image: Mean Squared Error (MSE)

The most basic and popular full-reference quality statistic is the mean squared error (MSE), which is calculated by averaging the squared intensity differences of distorted and reference picture pixels [57].
(30)$$MSE=\frac{1}{n}\sum_{i=1}^{N}{\left(O_{i}-I_{i}\right)}^{2}$$
where *N* is the number of pixels and $O_{i}$ and $I_{i}$ are the intensity values in the pixel *i*.

## 3. Results and Discussion

### 3.1. HLs Resolution

A *USAF Test 1951* has been used to determine the resolution of an optical system to assess how closely an object and its image resemble one another when holographic lenses are utilized. The test consists of a succession of groups of three bars repeated in a pattern. They typically fall within the 0.25 to 228 lp/mm range. Each group has six components. The group is identified by a Group Number (−2, −1, 0, 1, 2, etc.), which is equivalent to 2 raised to the power of the first element’s spatial frequency. Each element in the group is the sixth root of two units smaller than the element before it. The resolution of the system can be calculated using the equation or by reading the group and element number of the smallest pattern the optical system can resolve (using the condition of resolving at least 2 lines).
(31)$$Resolution=2^{Group+(Element-1)/6}$$

Convolution simulations of the USAF test with the ASF obtained by different methods were carried out.

#### 3.1.1. Simulated Convolution for ASF Obtained with the HS Wavefront Sensor

Figure 5 shows simulations of the convolution images between the USAF 1951 object test and the ASF obtained with the HS wavefront sensor.

#### 3.1.2. Simulated Convolution for $\left|ASF\right|$ Obtained with the CCD Sensor

Figure 6 shows simulations of the convolution images between the USAF 1951 object test and the ASF obtained with the CCD sensor.

#### 3.1.3. USAF Experimental Test Image Obtained with the CCD Sensor

Figure 7 shows the images of the experimental USAF object test obtained with the setup CCD sensor.

Table 3 shows the HL resolutions obtained in each of the methods. It can be seen that the best results were for the ASF obtained with the HS wavefront sensor (Figure 5). Secondly, good results were obtained for the direct experimental method (Figure 7). Finally, the worst results were obtained for the ASF obtained from the CCD sensor (Figure 6). It can be seen that the calculation of the resolution of HLs working with coherent light is not a simple task. The fact of approximating |ASF(x,y)|≈ASF(x,y) makes that part of the information is lost, which negatively affects the HLs resolution calculation.

In addition, to compare both methods, Table 4 shows the theoretical cut-off and Nyquist frequencies for the CCD and HS wavefront sensor. Previous works analysed both sensors in more detail [26,30].

To see the resolution of the HLs in more intuitive cases, simulations of the convolutions of the ASFs obtained with both methods (HS and CCD sensor) have been represented with the logo of the University of Alicante as an object (Figure 8). It can be seen that the image obtained for the HS sensor is sharper than the one obtained for the CCD sensor. Higher resolution can be seen in the case of the impulse response obtained with the HS sensor. In fact, it works as a low-pass filter smoothing the image.

### 3.2. MSE Metric

Finally, the MSE metric has been calculated to quantify the resolution of the HLs in terms of similarity between object and image. Figure 9 shows a scheme for the calculation of the MSE metric. Table 5 shows the results obtained for this metric. The difference between the object test and the image obtained from the convolution can be observed quantitatively. It can be seen that the results are consistent with the quality of the images obtained in each case.

Table 5 shows the results obtained for the MSE metric. It can be seen that the results obtained for the HS wavefront sensor are better than those obtained for the CCD sensor. This is in agreement with the previous results obtained for the resolution of the HLs.

To better understand the results, we can highlight some differences between the HS wavefront sensor and the CCD sensor. The HS wavefront sensor works with a CMOS (Complementary Metal Oxide Semiconductor) sensor which has some different characteristics than the CCD (Charge Coupled Device) sensor. We can compare both sensors using some important parameters. In terms of speed, CMOS is superior to CCD because all the processing is done inside the sensor itself, which offers higher speed. In the matter of noise, CCD is superior to CMOS. This is because the signal processing is done on an external chip, which may be better optimized to perform this function. In respect of blooming, CMOS sensors are superior to CCD. This phenomenon occurs when a pixel is saturated by the light incident on it and then begins to saturate the surrounding pixels. Finally, the dynamic range of the CCD sensor is twice that of the CMOS sensor. The dynamic range is the coefficient between the saturation of the pixels and the threshold below which they do not capture a signal. In this case, the CCD, being less sensitive, tolerates light extremes much better. Therefore, we have that the CMOS sensor has a higher speed and it’s better for the blooming phenomena, but the CCD sensor is better for the noise and dynamic range.

## 4. Conclusions

We have studied three different methods to analyze the resolution of HLs stored in a photopolymer material (Biophotopol). In this sense, we have improved the way to measure this resolution. The resolutions obtained with the CCD sensor using the convolution theorem are less reliable than those obtained with the Hartmann Shack sensor since part of the information is lost due to the way of obtaining the impulse response of the HL in this case. However, using the direct experimental method to calculate the resolution with the CCD sensor, similar results to those obtained with the HS sensor are obtained. The object-image similarity metric has also been analyzed to quantitatively evaluate the HL resolution. It can be observed that the best results have been for the negative asymmetric HLs reconstructed at 473 nm. It can be concluded that the HS wavefront sensor is a good tool to characterize the image quality and resolution of optical systems working with coherent light using the convolution theorem and the CCD sensor is a good tool to calculate the resolution by obtaining a direct experimental way the USAF test image. Moreover, the use of a low-toxicity photopolymer to store the holographic lenses allows us to obtain light, friendly, accurate, stable, and easy to fabricate systems compared with the traditional ones.

## Figures and Tables

**Figure 1 polymers-14-05426-f001:**
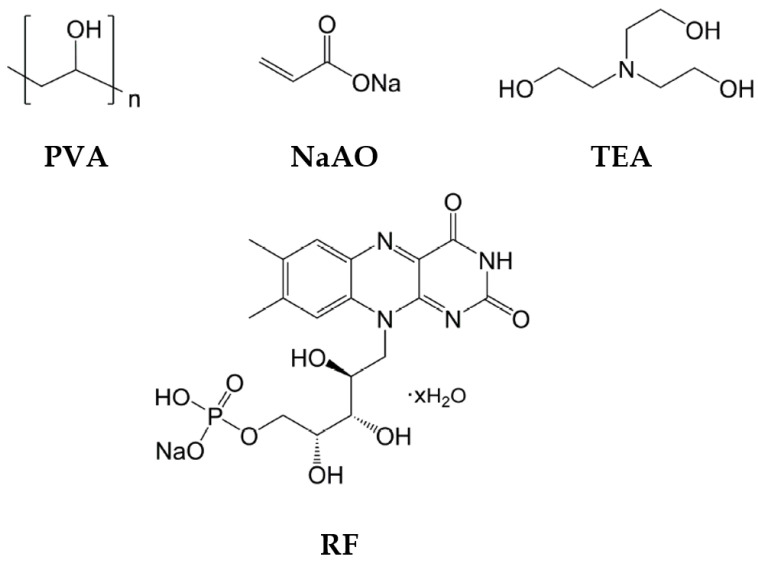
Biophotopol chemical structures of the prepolymer components. PVA: polyvinyl alcohol, NaAO: sodium acrylate, TEA: triethanolamine, RF: riboflavin 5’- monophosphate sodium salt.

**Figure 2 polymers-14-05426-f002:**
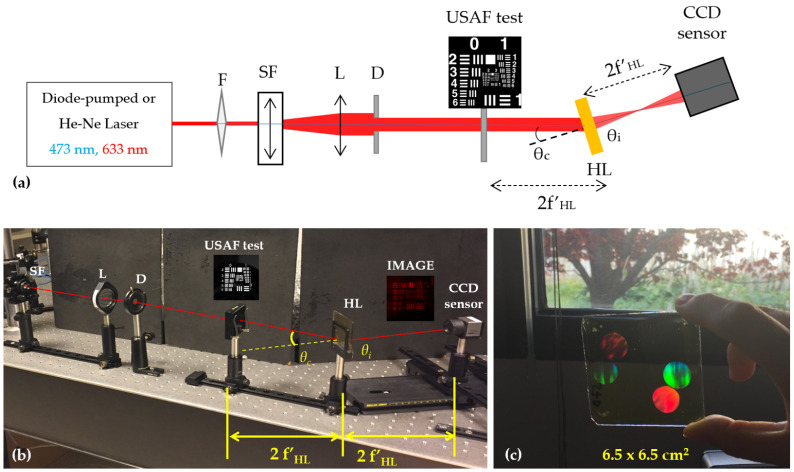
(**a**) Experimental setup for the HLs resolution evaluation. F: filter, SF: spatial filter, L: lens, D: diaphragm, HL: holographic lens, CCD Sensor: Charge Coupled Device. (**b**) Real photo of the experimental setup. (**c**) Picture of HL stored on the photopolymer layer. Video of HL stored on the photopolymer layer can be seen in Appendix A.

**Figure 3 polymers-14-05426-f003:**
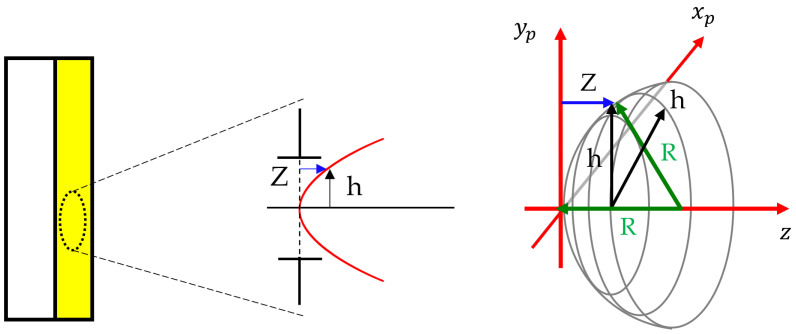
Geometry for obtaining the impulse response (ASF) of the HLs.

**Figure 4 polymers-14-05426-f004:**
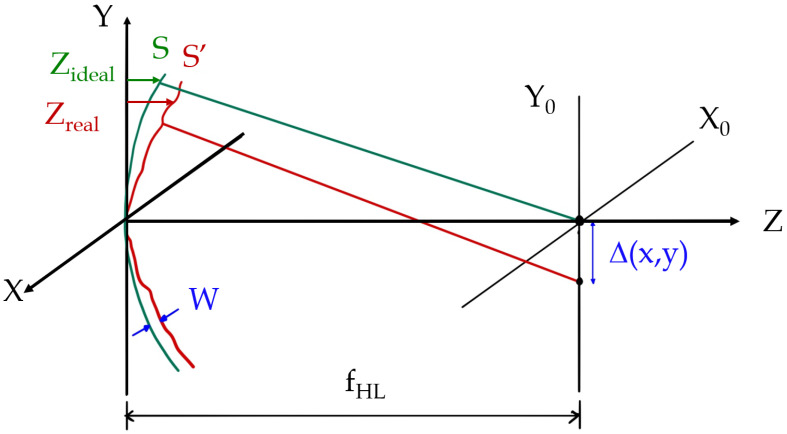
Geometry for the relation between the ideal sagittal, the real sagittal and the wave aberration function (*W*). The wave aberration (*W*) is also related to the ray aberration (Δ(x,y)).

**Figure 5 polymers-14-05426-f005:**
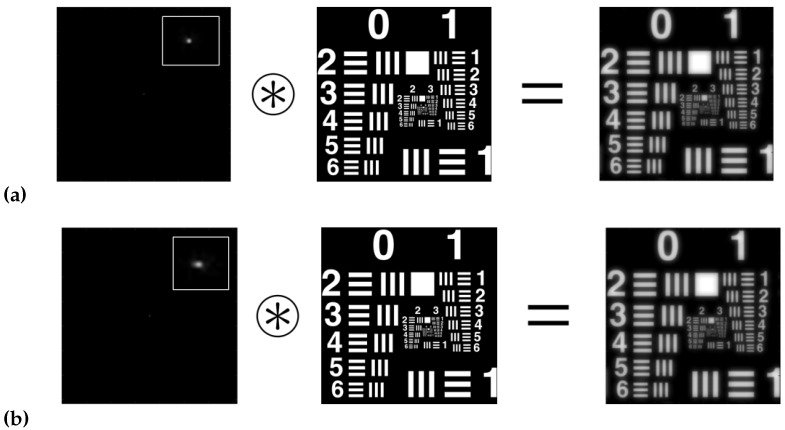
Simulated convolution for negative asymmetrical HLs, with ASF obtained with the HS wavefront sensor, reconstructed at (**a**) 473 nm and (**b**) 633 nm.

**Figure 6 polymers-14-05426-f006:**
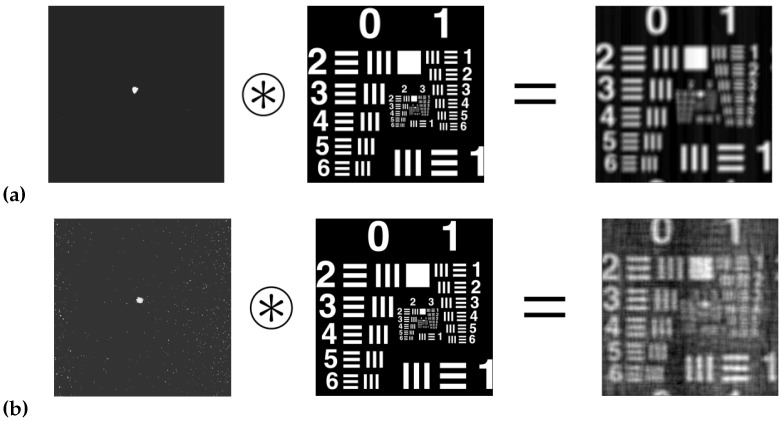
Simulated convolution for negative asymmetrical HLs, with ASF obtained with the CCD sensor, reconstructed at: (**a**) 473 nm, (**b**) 633 nm.

**Figure 7 polymers-14-05426-f007:**
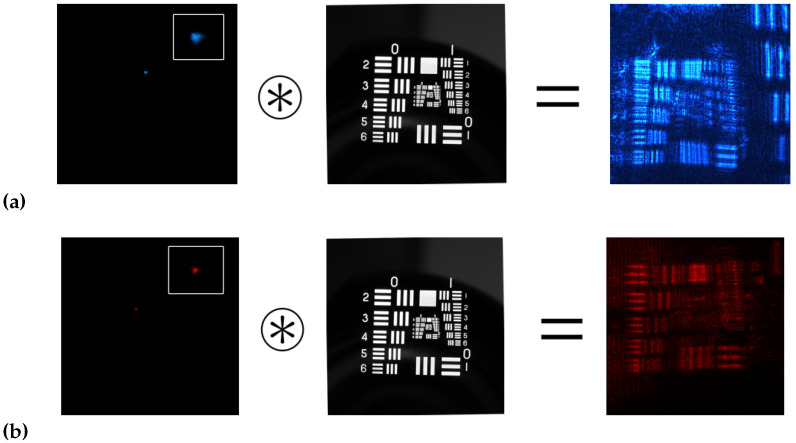
Image obtained with the CCD sensor illuminated at: (**a**) 473 nm, (**b**) 633 nm.

**Figure 8 polymers-14-05426-f008:**
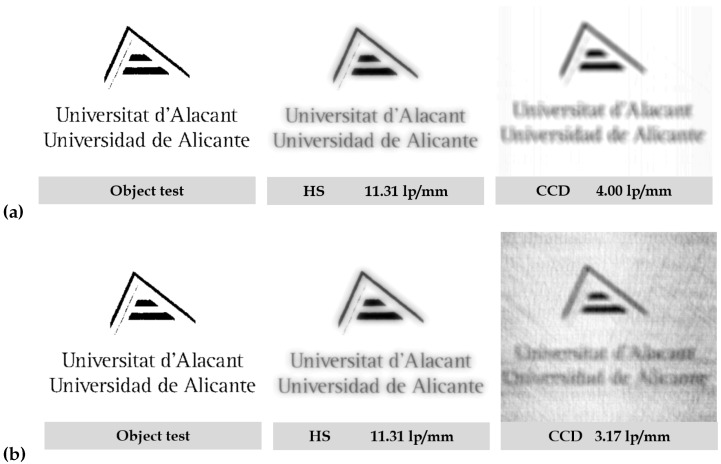
Convolution simulations of an object test (University of Alicante logo) with the ASFs obtained by the different methods (HS wavefront sensor and CCD sensor) using as reconstruction wavelength: (**a**) 473 nm and (**b**) 633 nm.

**Figure 9 polymers-14-05426-f009:**
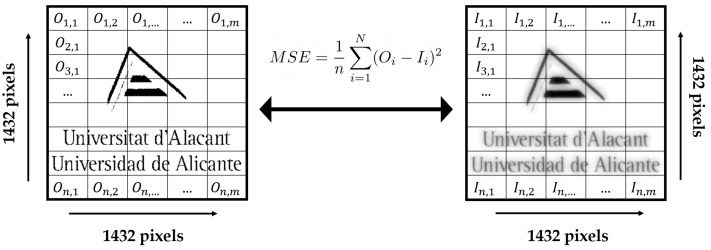
Convolution simulations of an object test (University of Alicante logo) with the ASFs obtained by the different methods: HS wavefront sensor and CCD sensor.

**Table 1 polymers-14-05426-t001:** Recording material composition quantities.

PVA	NaAO		TEA		RF	
(wt/V %) ^†^	(wt/V %)	(M) *	(wt/V %)	(M)	(wt/V %)	(M)
13.5	3.70	0.39	0.13	9.0 $\times 10^{-3}$	0.05	1.0 $\times 10^{-3}$

†: Weight by volume. *: Molarity.

**Table 2 polymers-14-05426-t002:** HL parameters for the reconstruction geometry.

λ (nm)	Angles and Focal Length	$\phi_{\mathrm{HL}}$ (mm)
	$\theta_{c}=33$°	
473	$\theta_{i}=0$°	12
	$f_{HL}^{\prime }=93$ mm	
	$\theta_{c}=46.8$°	
633	$\theta_{i}=0$°	12
	$f_{HL}^{\prime }=70$ mm	

**Table 3 polymers-14-05426-t003:** HLs resolution using the USAF test.

λ	USAF (Figure 5)	USAF (Figure 6)	Exp. USAF (Figure 7)
(nm)	(lp/mm)	(lp/mm)	(lp/mm)
473	11.31	4.00	10.08
633	11.31	3.17	8.98

**Table 4 polymers-14-05426-t004:** Theoretical cut-off and Nyquist frequencies for the CCD and HS wavefront sensor.

λ	F_cut_	F_Nyq_	F_Nyq_
(nm)	Theoretical	HS Sensor	CCD Sensor
	(lp/mm)	(lp/mm)	(lp/mm)
473	67.7	85.76	107.53
633	68.2	85.76	107.53

**Table 5 polymers-14-05426-t005:** Mean Squared Error (MSE) at 473 and 633 nm.

λ (nm)	MSE	MSE
(nm)	(HS Wavefront Sensor)	(CCD Sensor)
473	628.28	1853.70
633	629.31	3110.70

## Data Availability

Not applicable.

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
