# Peer review of "Holographic Lens Resolution Using the Convolution Theorem"

_polymers, 2022, doi:10.3390/polym14245426_

Round 1
Reviewer 1 Report
Dear Authors,
Manuscript polymers-2069806, entitle 'Holographic Lens Resolution using the Convolution Theorem' presents results on the resolution of holographic lenses (HLs) recorded on a low-toxicity recording material in asymmetrical recordings and with negative focal length. The manuscript is well written and organized. The conclusions are supported by the discussed findings.
The manuscript can be considered for publication after a good check of the text.
Minor Revision
Reviewer 2 Report
In the research of holographic display, large size, wide viewing angle and high resolution are very important. In the manuscript, the authors fabricated lens structures using polymer materials, multiple methods to improve the resolution of holographic displays are shown. This manuscript is a typical of cross research and application. But there are some details that need to be response in the next part.
1. At description of Materials and Methods section is not sufficient, after all, journal of Polymer is focus on material science. My personal suggestion is that to strengthen the description of this part.
2. In the main part of the manuscript, three different methods to analyze the resolution of HLs are shown.
1) A 473 nm diode-pumped laser and a 633 nm He-Ne laser were used to illuminate the HLs. How about green laser (532nm) and White laser?
2) In the study of holographic display, the view angle is very important. Please describe this part.
3. From my personal point of view, I’m looking forward to seeing more experimental results by changing the structure of the polymer lens.
